# Foliar Application of Humic Acid with Fe Supplement Improved Rice, Soybean, and Lettuce Iron Fortification

Sandeep Sharma [1], Neha Anand [1], Prem S. Bindraban [2] and Renu Pandey [1,*]

1 Division of Plant Physiology, ICAR-Indian Agriculture Research Institute, New Delhi 110012, India
2 International Fertilizer Development Center (IFDC), Muscle Shoals, AL 35662, USA
* Correspondence: renu_iari@rediffmail.com or renu_pphy@iari.res.in; Tel.: +91-11-2584-2815

**Abstract:** Iron (Fe) deficiency in humans, particularly in pregnant women and children, is caused by inadequate dietary Fe intake and is a global nutritional problem. Foliar fertilization is a cost-effective agronomic approach to increase Fe bioavailability in the human diet. We evaluated the effects of different Fe formulations (Fe-citrate, Fe-EDTA, $FePO_4$, nano-Fe oxide, and humic acid (HA) with and without Fe) on growth, yield, and Fe accumulation in the edible parts of rice, soybean, and lettuce crops. Rice and soybean received multiple sprays at different growth stages, i.e., tillering, anthesis, and grain filling in rice as well as flowering and pod filling in soybean, while lettuce received a single foliar spray. In rice and soybean, the seed Fe accumulation increased proportionally as the number of foliar sprays increased; however, the grain yield did not show this relationship. Among Fe treatments, HA+Fe was identified as the best treatment in terms of improving overall plant growth, yield, and Fe accumulation in the edible parts of all three crops. We found a significant positive correlation between the shoot/stover Fe content and the grain Fe content, but HA+Fe showed an opposite trend, i.e., minimal Fe retention in shoots/stovers and maximal increases in the seed Fe contents in both crops, suggesting better Fe mobilization efficiency from shoots to developing seeds. We strongly recommend that HA with Fe can be used as a foliar Fe fertilizer to improve the growth, yield, and Fe status in different crops.

**Keywords:** foliar spray; Fe use efficiency; Fe mobilization efficiency index; lettuce; nano-Fe; rice; soybean

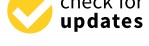



## 1. Introduction

Globally, more than two billion people suffer from iron (Fe) deficiency, and it is more common in children (6–24 months) and pregnant women from low- and middle-income countries [1]. The majority of staple foods lack the recommended daily intake of most micronutrients, particularly Fe, due to their inherent low contents of these elements [2]. Therefore, Fe fortification is necessary in the crops that are consumed by the majority of the world's population in order to address the Fe deficiency problem [3,4]. The best target crops for biofortification are rice (*Oryza sativa*) and soybean (*Glycine max*) because the former is a key staple food consumed by half the world's population and the latter is a major source of edible vegetable oil and protein for feed or food supplementation [1,5,6]. Furthermore, lettuce (*Lactuca sativa*), one of the most widely grown leafy vegetables, which is consumed fresh as salad, can be an optimal target for Fe biofortification [7]. Lettuce is not only a good source of minerals (potassium, magnesium, and calcium), vitamins (A, C, and E), and dietary antioxidants but is also known for its organoleptic properties [8]. The biofortification of lettuce with Fe would not only increase its nutritional and functional quality but would also provide an urgent and easy way to alleviate human Fe deficiency.

Although Fe is ubiquitous in the earth's crust, plant roots are unable to absorb it from soil due to its poor solubility and chemical instability. High soil pH and low soil moisture change the soil-available Fe ($Fe^{2+}$, soluble Fe) to plant-unavailable Fe, i.e., ferric

($Fe^{3+}$, insoluble Fe) [9]. Plant roots take up Fe via two strategies: Strategy I operates in dicotyledonous plants, which reduce $Fe^{3+}$ to $Fe^{2+}$ using the membrane-bound ferric chelate reductase (FCR) enzyme, and this reduced form enters the cells via the iron-regulated transporters (IRT). In strategy II, a chelation-based mechanism occurs in the Poaceae family, where the root releases a low-molecular-weight Fe-chelating compound called *phytosiderophore* (PS), which forms a complex with $Fe^{3+}$, leading to its efficient uptake by the yellow stripe 1 transporter (YS1) [10]. Soybean and lettuce use strategy I (the reduction strategy) to deal with Fe deficiency, while rice exhibits a combined strategy, that is, the traits of strategy II with some of the traits of strategy I, to acquire Fe from soil [10,11]. Therefore, the choice of Fe biofortification method should be such that it can be applied to different crops. To increase the bioavailability of Fe in human diets using a cost-efficient method is still a big challenge for the scientific community. The Fe biofortification of crops could be achieved by three approaches, namely conventional breeding, an agronomic approach, and genetic engineering. However, the key bottlenecks in the breeding approach are the yield factor, genotype–environment interactions, and the lack of genetic variability in modern cultivars, whereas the main challenges with genetically modified crops are consumer preferences and environmental safety [12]. On the other hand, agronomic biofortification involves applying nutrients to the soil [13] or directly to the foliage, which is a simple way to improve the Fe status of crops [14]. Soil-applied Fe fertilizers have a low success rate due to their low solubility and Fe fixation, so the best alternative to soil application for improving plant Fe status is foliar spray; however, soil application cannot be completely replaced by foliar application [15].

Increasing the Fe status of edible parts through foliar feeding is a short-term, targeted, cost-effective, and environmentally friendly strategy for crop biofortification. Effective foliar fertilization considers both the transport of Fe to the edible part and the efficiency of the penetration of foliar-applied Fe through plant foliage [16,17]. Several factors, including the physicochemical properties of the spray formulation (molecular size, solubility, pH, and surface tension), the morphological characteristics of the leaf (leaf shape, leaf size, and leaf surface chemistry), and environmental factors (light, temperature, wind, time of day, photoperiod, humidity, amount, and intensity of precipitation), influence the efficacy of foliar application [18,19]. Foliar-applied Fe successfully enters through stomatal pores or cuticular cracks and is absorbed by leaf epidermis, remobilized, and translocated into the sink (grain) via the phloem [15]. The higher agronomic effectiveness of the foliar feeding of different Fe forms for increasing grain Fe has been shown for rice [12,20], soybean [21–23], and wheat [14,23,24]. The efficacy of foliar application also depends on the number of foliar sprays and the growth stage at which the foliar is applied. The optimal time to apply foliar fertilizer is when a plant is in a transition phase, i.e., from the vegetative to the reproductive phase. The pre-anthesis stages are less tolerant to higher concentrations of foliar sprays than the post-anthesis stages [25]. Further, the Fe concentration in the spray solution may also differentially influence plant growth and the Fe accumulation in seeds. Armin et al. [26] observed no significant differences in yield attributes after foliar sprays of 4% and 6% nano-Fe solutions in wheat, while in soybean a lower dose (1%) of $FeSO_4$ was more effective in improving plant Fe content than the higher dose (2%) [21].

The accumulated evidence shows improved plant Fe deficiencies with foliar applications of chelate, inorganic, or nano-Fe compounds in various crops, but contradictory results have been obtained for foliar fertilization using these compounds [27]. The inorganic Fe salts exhibit increased efficiency of penetration and leaf regreening in comparison to Fe chelates. However, there is evidence that recommends the use of chelates, which increase Fe mobility [15,28,29]. Rodrigues-Lucena et al. [30] applied different Fe chelates (Fe-EDDS, Fe-EDTA, and Fe-IDHA) and complexes (citrate, humates, gluconate, glycoprotein, lignosulfonate, and polyamines) to soybean and concluded that chelates showed some advantages over complexes with respect to plant growth, Fe concentration, and SPAD values. Recent advances in innovative technology have increased the plant growth and yields of several crop species by applying nano-Fe fertilizers via leaves [31,32]. Because of

its diminutive size, Fe nano-oxide can enter plant cells through ion channels or aquaporins by binding to carrier proteins [33]. In the recent past, the foliar application of humic acid (HA), either alone or in combination with other nutrients, has been developed to promote plant growth by increasing photosynthesis, nutrient uptake, hormone activity, and antioxidant scavenging capability [34–37]. Although the available literature suggests the use of various Fe fertilizers to enhance growth and Fe biofortification, there is still no definite conclusion concerning the forms of Fe compounds suitable for various crop species. Therefore, we hypothesized that there may be a single Fe compound that could be used as a foliar fertilizer in different crop species for Fe biofortification. In this study, we investigated the effects of various foliar Fe compounds in crops belonging to different functional food groups, i.e., rice, soybean, and lettuce, grown in soil under non-Fe-deficient natural climatic conditions. The objectives were to find a suitable foliar Fe fertilizer that could be used to enhance the amount of Fe in edible parts by using the right Fe concentration and number of foliar sprays at the right growth stage.

## 2. Materials and Methods

### 2.1. Plant Material and Growth Conditions

Pot experiments were conducted with three different crop species, rice (variety MAS 946-1), soybean (variety DS-2614), and lettuce (Chinese yellow), under natural growing conditions in their respective seasons at the ICAR-Indian Agriculture Research Institute, New Delhi, India. The soil (0–30 cm) was collected from the experimental field, air-dried, and sieved through 5 mm mesh. A subsample of soil was analyzed for nutrients and physicochemical properties (Table S1). For each crop, the recommended dose of fertilizer (N:P:K kg ha$^{-1}$), 120:60:40, 10:60:40, and 50:30:30 for rice, soybean, and lettuce, respectively, was added to the soil and filled in pots (30 cm top diameter and 30 cm height). The amounts of urea, single superphosphate (SSP), and muriate of potash (MOP) were calculated according to the soil volume in each pot. For rice, the amounts of urea, SSP, and MOP applied per pot were 1.73 g, 2.50 g, and 0.45 g, respectively. In soybean, 0.15 g pot$^{-1}$ of urea was applied, while the SSP and MOP amounts were the same as those for rice. For lettuce, the quantities of urea, SSP, and MOP applied in each pot were 0.717 g, 1.25 g, and 0.33 g, respectively. All fertilizers were applied as basal doses in all crops, except urea in rice, which was applied in split doses, i.e., 50% as basal, 25% at tillering, and 25% at anthesis. For soybean, seeds were sown directly in the pot, while for rice and lettuce a nursery was prepared, followed by the transplanting of 25-day-old seedlings to the pots. Two healthy plants were retained per pot for each crop. Plants were irrigated as required with tap water.

### 2.2. Foliar Treatments

To determine the optimal concentrations of various Fe compounds, preliminary experiments were carried out for all crops in the glasshouse at the National Phytotron Facility, ICAR-Indian Agriculture Research Institute, New Delhi (data not presented). The details of each foliar treatment applied to these crops are presented in Table 1. The spray solutions for each Fe compound were made with distilled water and 0.1% Triton X100 (Sigma 9036-19-5) as a surfactant. The pH values of all spray solutions were adjusted to 6.0 using 1 N KOH/HCl. Before spraying, the rice and soybean plants were divided into different sets, and each set was different in terms of the number of sprays applied at different growth stages. For experiment 1, the rice plants were divided into five sets with an equal number of pots. The set 1, set 2, and set 3 plants received single sprays in the tillering, anthesis, and grain-filling stages, respectively. The set 4 plants were sprayed twice in the anthesis and grain-filling stages, while the set 5 plants received three sprays (in all stages) (Figure 1). For experiment 2, the soybean plants were divided into three sets. The set 1 and set 2 plants received single sprays at flowering (50 days after sowing, when plants began to bloom) and pod filling (75 days after sowing, at the beginning of seed setting), respectively, while the set 3 plants were sprayed twice (at both flowering and pod filling) (Figure 1). In

experiment 3, lettuce plants were sprayed only once with different Fe formulations 30 days after transplanting. Before spraying in all crops, the base of each plant was covered with polythene to avoid the dripping of excess spray solution from the foliage into the soil. Further, the plants were sprayed with an Fe solution in the forenoon (before 10 am) because, during this time, humidity is higher and leaves remain in a state of full turgor, which leads to the maximum absorption of nutrients from the foliage.

**Table 1.** Details of Fe formulations applied as foliar spray on rice, soybean, and lettuce in different growth stages.

| Crop | Fe Formulations (Source) | Concentration | Abbreviation Used |
|---|---|---|---|
| Rice | Fe-Citrate (Sigma 3522-50-7) | 2 mM and 4 mM | Fe-Cit-2 and Fe-Cit-4 |
| | Fe-EDTA (Sigma 149022-26-4) | 2 mM | Fe-EDTA-2 |
| | Fe-phosphate (Sigma 13463-10-0) | 4 mM | FeP-4 |
| | Humic acid (Sigma 1415-93-6) + ferric chloride (Sigma 7705-08-0) | 25 mg L$^{-1}$ + 2 mM | HA+Fe |
| | Nano Fe (Sigma 1309-37-1) (particle size < 50 nm) | 2 mM and 4 mM | nano-Fe-2 and nano-Fe-4 |
| Soybean | Fe-Citrate (Sigma 3522-50-7) | 4 mM | Fe-Cit-4 |
| | Fe-phosphate (Sigma 13463-10-0) | 4 mM | FeP-4 |
| | Humic acid (Sigma 1415-93-6) + ferric chloride (Sigma 7705-08-0) | 25 mg L$^{-1}$ + 2 mM | HA+Fe |
| | Humic acid (Sigma 1415-93-6) | 50 mg L$^{-1}$ | HA |
| | Nano Fe (Sigma 1309-37-1) | 4 mM | nano-Fe-4 |
| Lettuce | Fe-Citrate (Sigma 3522-50-7) | 2 mM | Fe-Cit-2 |
| | Fe-phosphate (Sigma 13463-10-0) | 2 mM | FeP-2 |
| | Humic acid (Sigma 1415-93-6) + ferric chloride (Sigma 7705-08-0) | 25 mg L$^{-1}$ + 2 mM | HA+Fe |
| Control | Deionized water | - | Control |

**Figure 1.** The layout of foliar spraying schedules at different growth stages. The green shade represents the stage of growth in each crop at which foliar application was performed in different sets.

*2.3. Physiological Traits and Tissue Nutrient Analysis*

In rice and soybean, observations on growth and yield attributes were recorded at the physiological maturity stage, including the total aboveground biomass, panicle number plant$^{-1}$, filled-grain percentages, test weight (1000 seed weight), grain weight panicle$^{-1}$, and grain yield plant$^{-1}$ in rice. Likewise, the pod number, pod weight, seed weight pod$^{-1}$, test weight (100 seed weight), and seed yield plant$^{-1}$ were recorded in soybean. For Fe estimation in tissues, samples were digested using a di-acid mixture of nitric (Sigma 231-714-2) and perchloric acids (Sigma 7601-90-3) in the ratio of 9:4, following a standard protocol [38]. Fe (%) estimation was carried out using an inductively coupled plasma optical emission spectrometer (5110 ICP-OES, Agilent Technologies, Singapore). The Fe content or uptake was calculated by multiplying the Fe concentration with the dry weight of the respective plant part and was expressed as mg plant$^{-1}$. Other traits, such as the Fe use efficiency (FeUE), Fe harvest index (FeHI), and Fe mobilization efficiency index (FeMEI), were computed using the following formulae [39,40]:

$$\text{Fe use efficiency (FeUE)} = \frac{\text{Total grain yield}}{\text{Total above ground Fe uptake at harvest}}$$

$$\text{Fe harvest index (FeHI)} = \frac{\text{Total Fe uptake by grain}}{\text{Total above ground Fe uptake at harvest}}$$

$$\text{Fe mobilization efficiency index (FeMEI)} = \frac{\text{Fe concentration in grain (mg/kg)}}{\text{Fe concentration in straw (mg/kg)}}$$

In lettuce, the observations were recorded on the sixth day after foliar application. The fresh and dry weights of shoots, leaf area, and Fe concentration in young leaves and fully expanded leaves were measured/estimated. The dry matter percentage was calculated by dividing the shoot dry weight by the shoot fresh weight. The total leaf area was measured by a leaf area meter (LiCor-3000). The leaf area ratio, which represents the leafiness of the plant, was calculated as the ratio of the total leaf area to the total leaf dry weight and was expressed as cm$^2$ g$^{-1}$ [41]. The Fe estimation in leaves was performed following the same protocol as mentioned above for rice and soybean in Section 2.3.

*2.4. Statistical Analysis*

The experiments on rice and soybean were laid out in a completely randomized design (CRD) with two factors, foliar treatments (T) and growth stages (S), while the experiment on lettuce was carried out in a CRD with one factor, foliar treatments (T). The number of replications for each treatment was five, and one pot served as one replicate, with two plants per pot in all three experiments. For basic statistical analysis and ANOVA, an MS-DOS based statistical software package, AGRES version 3.01, was used [42]. To quantify the association between traits, Pearson's correlation coefficient and a linear regression were calculated using MS Excel 2016. Graphs were made using GraphPad Prism version 6.0 (GraphPad Software, La Jolla, CA, USA).

**3. Results**

*3.1. Influence of Foliar Fe Application on Biomass Accumulation and Yield Attributes*

In rice, the above-ground biomass (AGB) and yield attributes significantly ($p < 0.01$) increased with all foliar treatments (T) compared to control in all sets, while between sets (S) the averages over treatments and the interaction of both (S × T) were significant ($p < 0.05$) for most of the traits, the except test weight, filled-grain percentage, and grain weight panicle$^{-1}$ (Figure 2; Table S2). Among different sets, averaged over all treatments, the maximum increment in AGB was recorded in set 4 (67.2%) compared to the control, while the lowest was recorded in set 1 (33.6%) (Figure 2a). Compared to control, the minimum increase in biomass was recorded by nano-Fe-4 in set 1, while the maximum increase (>2-fold) was recorded with FeP-4 and Fe-Cit-2 in sets 2 and 4, respectively. In all sets, with the exception of set 1, spraying HA+Fe increased AGB consistently by more than 72%

compared to control. Among sets, the mean of treatments for grain yield was highest in sets 2 and 3 (>50% compared to control), while the panicle numbers were highest in sets 4 and 5 (>10% compared to control) (Figure 2b,f). All treatments, except nano-Fe-2 (set 4), increased the number of filled grains panicle$^{-1}$ compared to control, with Fe-Cit-2 (set 1) being the highest. However, Fe-Cit-2 had the most consistent effect on the grain-filling percentage, and its application increased grain filling by more than 20% compared to control in all sets (Figure 2c). The weights of individual panicles also increased significantly with all foliar treatments compared to control, with the maximum increases recorded with FeP-4 (set 3) and Fe-Cit-2 (set 4). The maximum increase (>2-fold) in the total grain weight was noted with HA+Fe and FeP-4 application in set 3, while minimum was recorded with the higher concentration of nano-Fe in set 1. The foliar application of HA+Fe showed consistent results compared to other treatments in terms of increasing the total grain weight, which resulted in more than >70% increases in all sets in comparison to control.

In soybean, Fe treatments (T), different sets (S), and their interaction (T × S) significantly influenced ($p < 0.01$) the AGB accumulation and yield attributes (Figure 3; Table S3). Among sets, the maximum AGB was recorded in set 3 (>27%), while no significant differences were recorded in set 1 and set 2 compared to control. Among Fe compounds, the maximum increase (>2-fold) in AGB was observed with FeP-4 and Fe-Cit-2 in sets 2 and 4, respectively, while nano-Fe-4 (set 1) showed the smallest increment (Figure 3a). All Fe formulations significantly increased most of the yield traits in all three sets, except the seed weight pod$^{-1}$ (Figure 3b–f). Among the sets, set 3 showed the maximum increases in all yield traits, ranging from 29% to 129% compared to control, with the exception of the seed weight pod$^{-1}$. Among Fe compounds, the maximum increases in the pod weight (>2.4-fold), pod number (>3.0-fold), and total seed weight (>3.4-fold) were recorded with the foliar application of HA+Fe compared to control in set 3. Furthermore, the seed weight pod$^{-1}$, which is one of the major yield-determining traits, along with the pod number in soybean, increased in all sets by 10.0% (set 1), 22.5% (set 2), and 18.1% (set 3) with the foliar application of HA+Fe compared to control. Although nano-Fe-4 resulted in the maximum increase (49%) in seed weight pod$^{-1}$ (set 1), the total seed yield plant$^{-1}$ was poor, which was due to a considerable decrease (>17) in the pod number plant$^{-1}$.

In lettuce, the foliar application of different Fe formulations significantly ($p < 0.05$) improved the overall plant growth by increasing the biomass and leaf area (Figure 4). All Fe treatments significantly increased the shoot dry weight compared to control, with the maximum recorded in HA+Fe (51%) (Figure 4a), whereas the maximum increase in the shoot fresh weight was recorded with Fe-Cit-2 (29%), followed by HA+Fe (26%) (data not presented). The maximum increase in the dry matter percentage was obtained with HA+Fe (29%) compared to the control (Figure 4b). Similarly, the leaf area also increased with all Fe treatments. However, the application of HA+Fe and Fe-Cit-2 resulted in >70% increases compared to control (Figure 4c). The leaf area ratio showed no significant differences among the Fe compounds (Figure 4d).

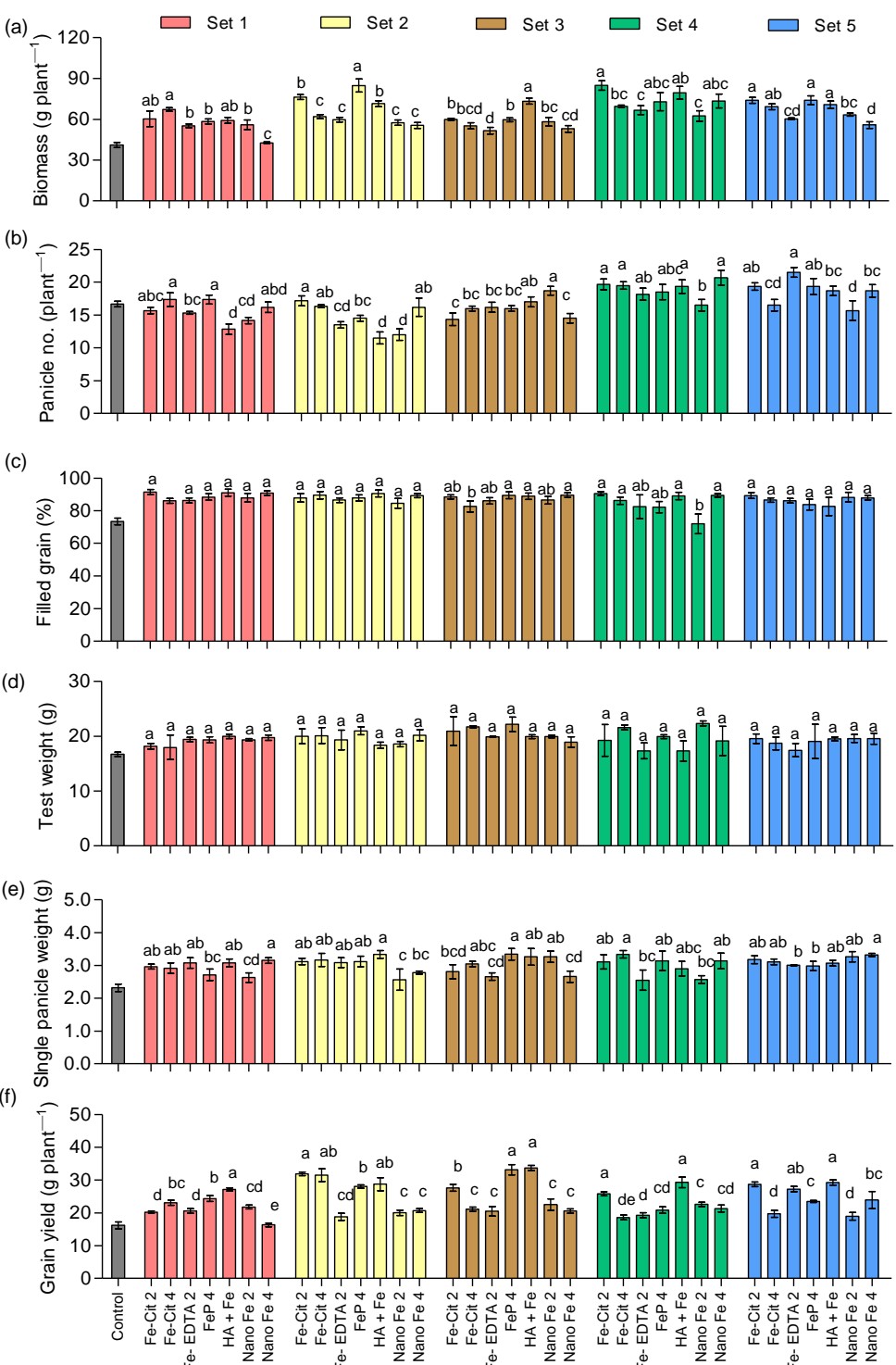

**Figure 2.** Influence of foliar application of Fe compounds on (**a**) total aboveground biomass, (**b**) number of panicles, (**c**) filled-grain percentage, (**d**) test weight, (**e**) single-panicle weight, and (**f**) grain yield of rice. Data correspond to means ± SEm (*n* = 5). The plants of set 1, set 2, and set 3 received single sprays in the tillering, anthesis, and grain-filling stages, respectively, while set 4 plants were sprayed twice (anthesis and grain-filling stages) and set 5 plants were sprayed in all three growth stages. Each set was analyzed separately using a one-way ANOVA, and the least significant difference was calculated. Means with the same letter were not significantly different at *p* < 0.05. Details of the two-way ANOVA are presented in Table S2.

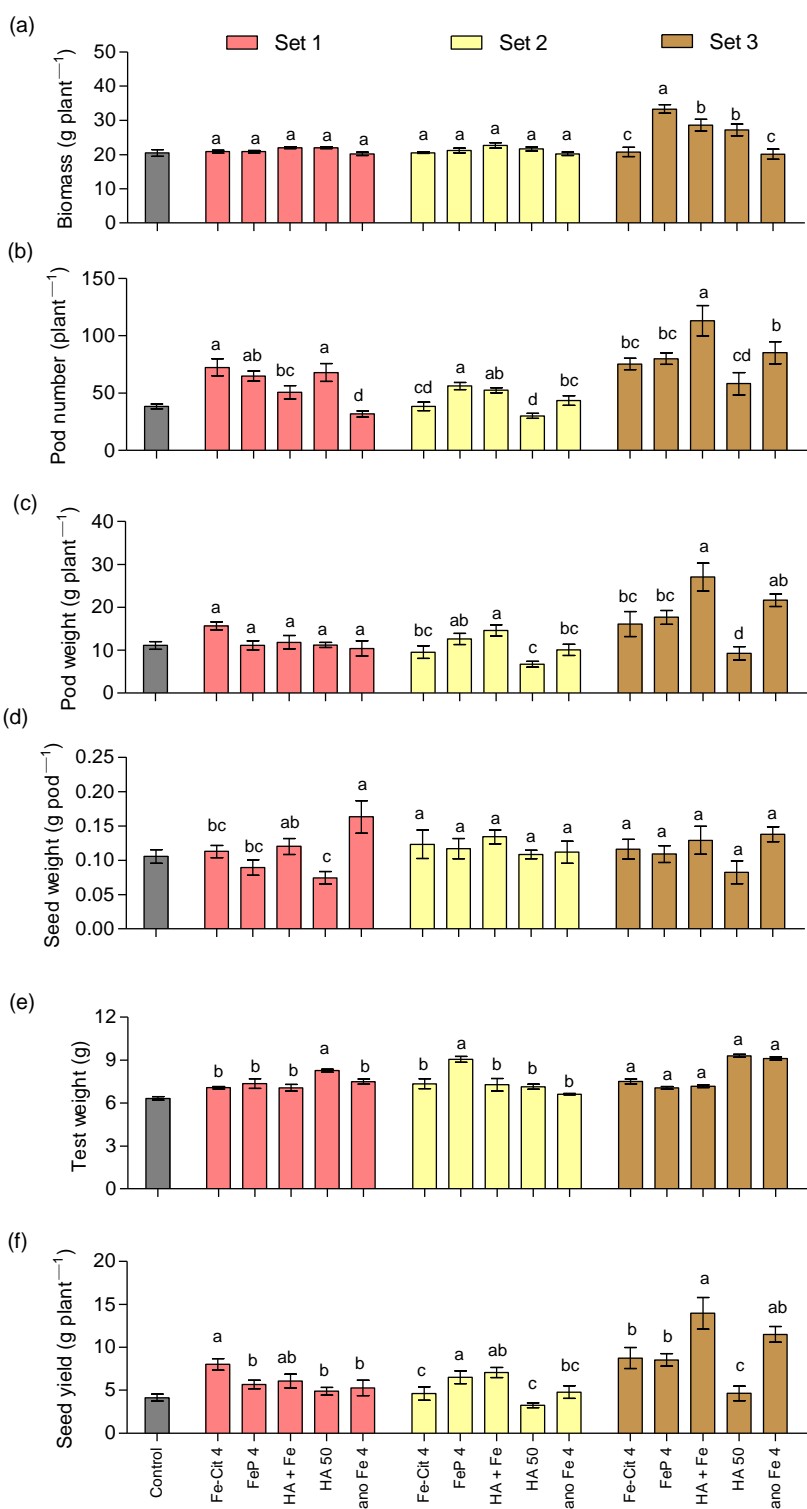

**Figure 3.** Influence of foliar application of Fe compounds on (**a**) total aboveground biomass, (**b**) number of pods, (**c**) pod weight, (**d**) seed weight pod$^{-1}$, (**e**) test weight, and (**f**) total seed yield of soybean. Data correspond to means ± SEm (*n* = 5). The plants of set 1 and set 2 received single sprays in the flowering and pod-filling stages, respectively, while set 3 plants were sprayed in both stages. Each set was analyzed separately using a one-way ANOVA, and the least significant difference was calculated. Means with the same letter were not significantly different at *p* < 0.05. Details of the two-way ANOVA are presented in Table S2.

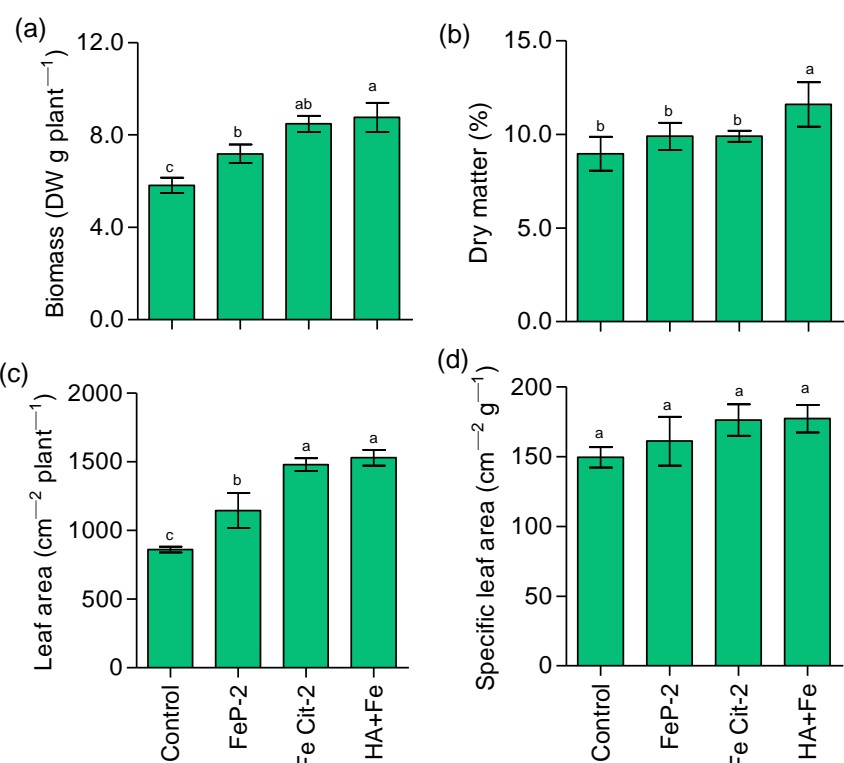

**Figure 4.** Influence of foliar application of Fe compounds on (**a**) total aboveground biomass, (**b**) dry matter, (**c**) total leaf area, and (**d**) leaf area ratio of lettuce. Data correspond to means ± SEm (*n* = 5). Means with the same letter were not significantly different at *p* < 0.05.

Among all three crops, the foliar application of HA+Fe showed the most promising results with respect to improving overall growth and yield. Its application increased the AGB in all three crops and significantly increased the grain yields in rice and soybean in all sets.

### 3.2. Influence of Foliar Fe Application on Fe Concentration and Uptake in Rice, Soybean, and Lettuce

In rice, significant (*p* < 0.01) effects of Fe treatments (T) and sets (S) were observed on the Fe contents and concentrations in grain and straw (Figure 5b–f; Table S2). Among different sets, the maximum increases in the Fe contents in grain and straw were observed when spraying was performed in all three stages (set 5), while the lowest was observed in set 1, with a single spray in the tillering stage (Figure 5). Likewise, the maximum increases (>40%) in the Fe concentrations in grain and straw were observed in set 5 in comparison to control, while the smallest increases (14%) were observed in set 1, though they were significant (Table S4). Compared to control, all Fe formulations in all sets significantly improved the Fe status of rice plants. Among the treatments, nano-Fe (4 mM) showed the maximum increases in the straw and grain Fe concentrations by 66.8% and 84.5%, respectively, in set 5. With respect to the grain Fe content, the foliar application of HA+Fe in each set resulted in the maximum increase, with increments ranging from 93% in set 1 to 178% in set 5 compared to control. The Fe compounds increased the straw Fe uptake by >2.1-fold in set 4 and set 5, with the highest being noted in Fe-Cit (4 mM) and FeP in set 5 compared to control (Figure 5b–f).

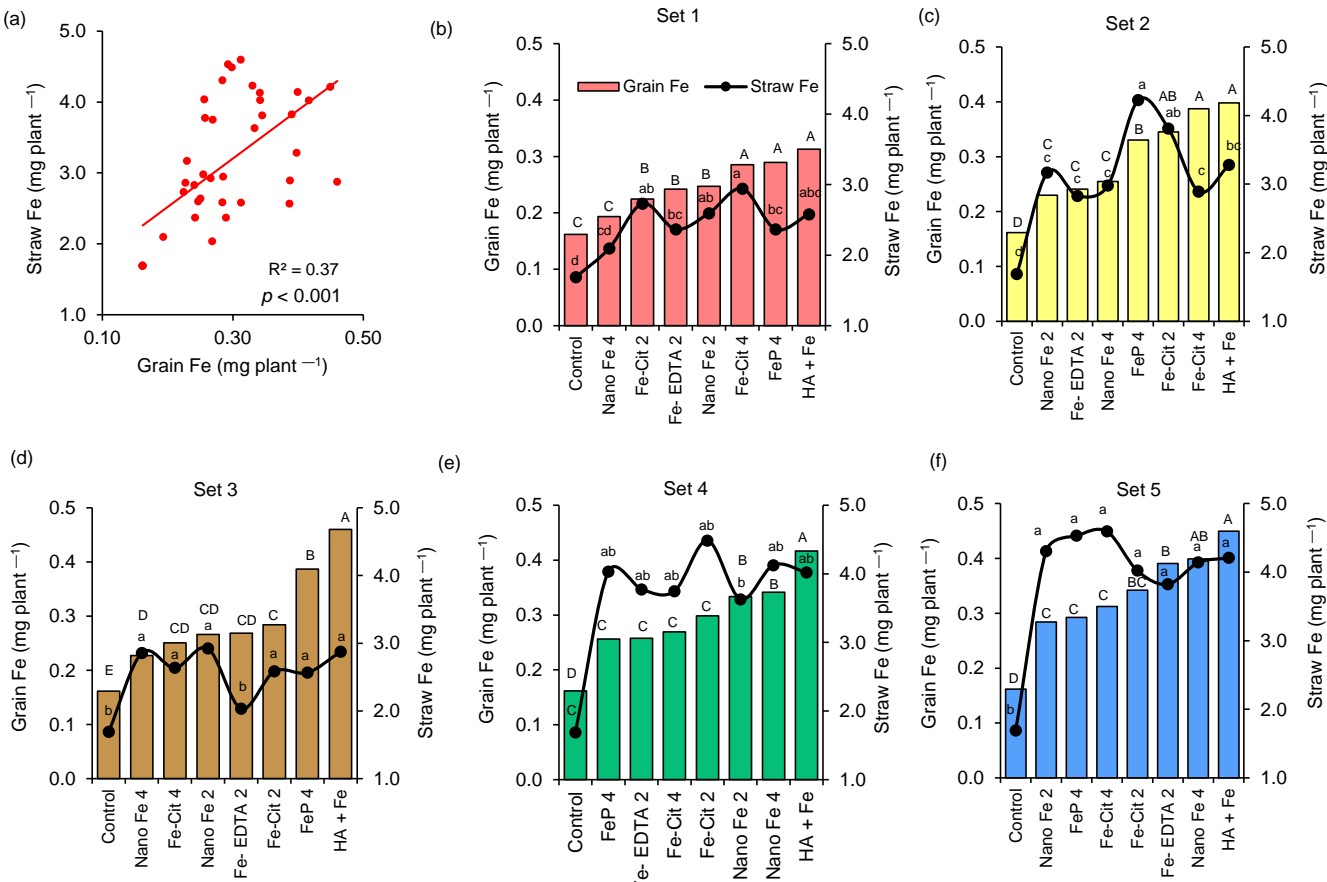

**Figure 5.** Influence of foliar application of Fe compounds on (**a**) a linear regression between grain Fe and straw Fe contents in rice (values are means of all 5 sets) and (**b**–**f**) grain Fe contents and straw Fe contents in different sets in rice. Data correspond to means ± SEm (*n* = 5). The plants of set 1, set 2, and set 3 received single sprays in the tillering, anthesis, and grain-filling stages, respectively, while set 4 plants were sprayed twice (anthesis and grain-filling stages) and set 5 plants were sprayed in all growth stages. Each set was analyzed separately using a one-way ANOVA, and the least significant difference was calculated. Means with the same letter (uppercase for grain Fe content and lowercase for straw Fe content) were not significantly different at *p* < 0.05.

In soybean crops, the seed and stover Fe concentrations and contents were significantly (*p* < 0.05) influenced by the Fe formulations (T), the number of foliar sprays in different growth stages (S), and their interactions (S × T) (Figure 6, Table S3). Among sets, the highest seed Fe concentration and content were achieved in set 3, i.e., single sprays in the both flowering and pod-filling stages, while the smallest increase was noted with a single spray during pod filling (set 2). Similarly, the maximum increases in the stover Fe concentration and content were observed in set 3, while the smallest was recorded in set 1 compared to control. In set 3, spraying all Fe formulations, except nano-Fe-4, resulted in >2-fold increases in the stover Fe content compared to control. With respect to the seed Fe content, the foliar spray of HA+Fe and nano-Fe-4 showed significant increases in all sets, with the highest being recorded in set 3 (>3.4-fold). However, the seed Fe concentration was highest in set 3, with Fe-Cit-4 (39.1%), followed by nano-Fe-4 (25.7%), while the lowest was observed in the HA 50 treatment.

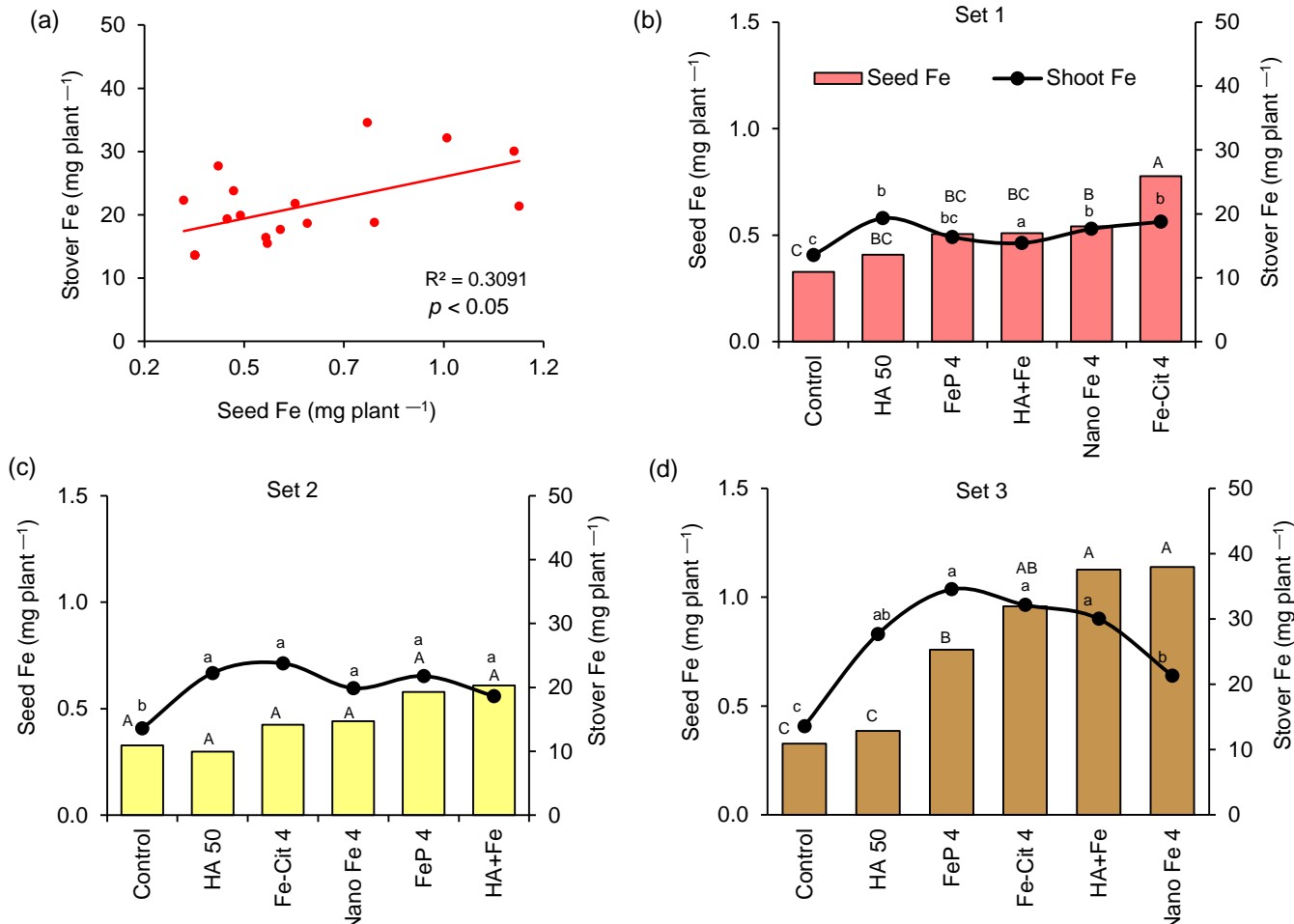

**Figure 6.** Influence of foliar application of Fe compounds on (**a**) a linear regression between the seed Fe and straw Fe contents in soybean (values are means of all 3 sets) and (**b–d**) seed Fe contents and stover Fe contents. Data correspond to means ± Sem (*n* = 5). The plants of set 1 and set 2 received single sprays in the flowering and pod-filling stages, respectively, while set 3 plants were sprayed in both stages. Each set was analyzed separately using a one-way ANOVA, and the least significant difference was calculated. Means with same letter (uppercase for grain Fe content and lowercase for straw Fe content) were not significantly different at $p < 0.05$.

In lettuce, the foliar application of all Fe formulations significantly ($p < 0.05$) improved the Fe status in leaves (Figure 7). Among treatments, the Fe concentrations in both young and fully expended leaves were the highest with HA+Fe, followed by Fe-Cit. Similarly, the maximum increase in the shoot Fe content was observed with HA+Fe (90.9%), followed by Fe-Cit-2 (66.1%) and FeP-2 (38.5%), compared to control.

Overall, the seed Fe concentrations and contents in both rice and soybean were directly proportional to the number of sprays of the Fe formulation, and in both crops the maximum seed Fe status was achieved with the maximum number of sprays. Among Fe formulations, HA+Fe was identified as the best treatment for Fe biofortification in all three crops.

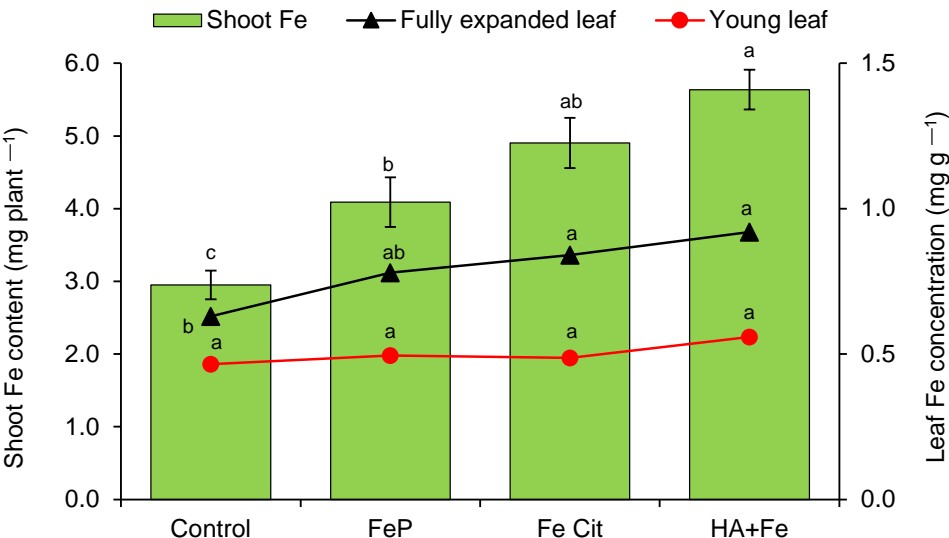

**Figure 7.** Influence of foliar application of Fe compounds on shoot Fe content and Fe concentration in young leaves and fully expanded leaves of lettuce. Data correspond to means ± SEm (*n* = 5). Means with same letter were not significantly different at *p* < 0.05.

### 3.3. Influence of Foliar Fe Application on Fe Use Efficiency and Indices in Rice and Soybean

In both rice and soybean crops, the Fe harvest index (FeHI), Fe use efficiency (FeUE), and Fe mobilization efficiency index (FeMEI) were significantly (*p* < 0.01) influenced by the Fe compounds and the number of foliar sprays in different sets (Figures 8 and 9; Tables S2 and S3). In rice, a consistent increase in FeHI was observed with HA+Fe application in all sets, with >25% increases in the first three sets (set 1, 2, and 3). However, most of the Fe formulations resulted in significant reductions in FeHI, and the effect was prominent in sets 4 and 5 (Figure 8a). The FeUE, described as the amount of grain produced per unit of Fe in a plant, increased with the application of FeP-4 and HA+Fe in set 3, while all Fe compounds showed considerable decreases (Figure 8b). Significant reductions in FeUE in sets 4 and 5 suggest that a greater number of Fe sprays does not increase the grain yield but rather increases the Fe content in shoots. Regarding FeMEI, representing the mobilization of Fe from the source (foliage) to the sink (grain), significant increases were noted in all sets with the foliar application of HA+Fe (Figure 8c). Among the sets, the maximum increase in FeMEI was recorded in set 3 (52%), followed by set 2 (38%). Besides HA+Fe, significant increases in FeMEI were also recorded with the application of Fe-EDTA-2 in the first three sets.

In soybean crops, HA+Fe showed a significant increase in FeHI compared to control in all sets except set 1; however, among the treatments, the maximum FeHI was recorded with nano-Fe-4 (>2.0-fold) in set 3 (Figure 9a). Between the sets, the maximum increase (>26%) in FeHI was noted in set 3 (sprayed at both flowering and pod filling) compared to control. The FeUE was also the highest in set 3 compared to control, while the maximum increase in FeMEI occurred in set 2 (Figure 9b,c). Among the Fe treatments, nano-Fe-4 resulted in the maximum increase in FeUE and FeMEI in set 3 and set 2, respectively, in comparison to control, while HA+Fe showed consistent increases in both traits in all sets except set 1.

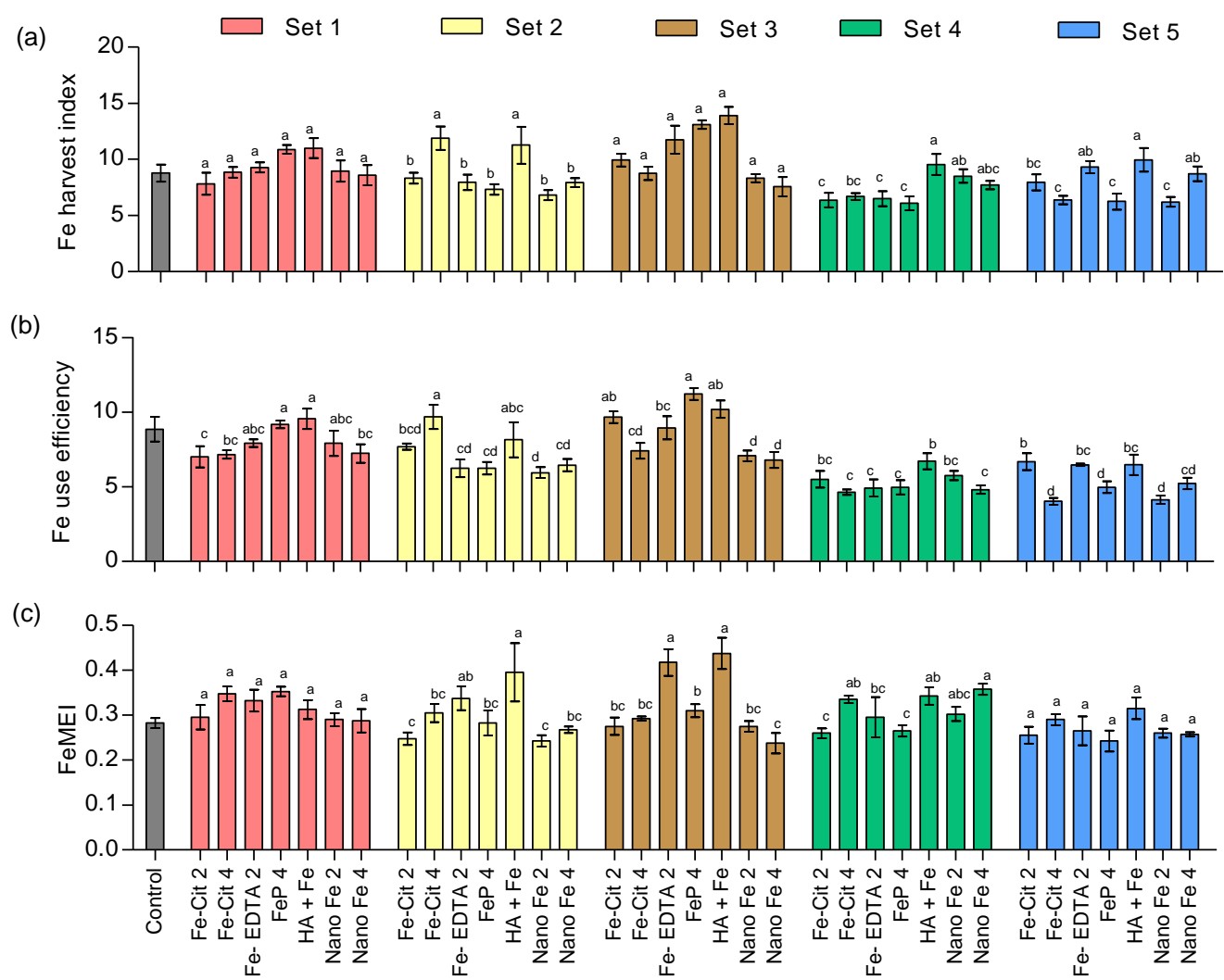

**Figure 8.** Influence of foliar application of Fe compounds on Fe indices in rice: (**a**) Fe harvest index, (**b**) Fe use efficiency, and (**c**) Fe mobilization efficiency index (FeMEI). Data correspond to means ± SEm (*n* = 5). The plants of set 1, set 2, and set 3 received single sprays in the tillering, anthesis, and grain-filling stages, respectively, while set 4 plants were sprayed twice (anthesis and grain-filling stages) and set 5 plants were sprayed in all three growth stages. Each set was analyzed separately using a one-way ANOVA, and the least significant difference was calculated. Means with same letter were not significantly different at $p < 0.05$. Details of the two-way ANOVA are presented in Table S2.

### 3.4. Association between Fe Contents in Grain and Straw and FeMEI

The grain Fe contents showed a significant positive association with the straw/stover Fe contents in rice ($p < 0.001$; $R^2 = 0.37$) and soybean ($p < 0.05$; $R^2 = 0.31$) (Figures 5a and 6a). However, when the performances of individual Fe formulations were compared in each set, this association did not hold true; rather, we found the opposite trend. For example, in rice, HA+Fe was the best-performing Fe formulation, which resulted in the highest grain Fe content in all sets (Figure 5b–f), but the association between the Fe content in straw and grain Fe was negative. This indicates that HA helped in the remobilization of Fe from straw towards grain, which was clearly visible, as the Fe content in straw was reduced but increased in grains. However, in set 3 (Figure 5d), with a single spray in the grain-filling stage, it was observed that most of the Fe formulations exhibited negative relationships between the Fe contents in straw and grain, indicating better Fe mobilization towards grains. Similarly, in soybean, in each set an Fe-formulation-dependent association between the seed Fe and stover Fe contents was observed (Figure 6b–d). Only set 3 plants (Figure 6d) in soybean sprayed twice with nano-Fe-4 and HA+Fe showed significant negative associations

between stover and seed Fe contents, indicating a variable response of the Fe formulations. Interestingly, we observed a crop-dependent association between FeMEl and the grain Fe content. In rice, a linear regression of FeMEI with the grain Fe content showed a significant positive relationship in all sets (set 1, $p < 0.001$; set 2, $p < 0.05$; sets 3 and 4, $p < 0.01$) except set 5 ($p = 0.406$), while in soybean the relation between FeMEI and the grain content was not significant in all sets.

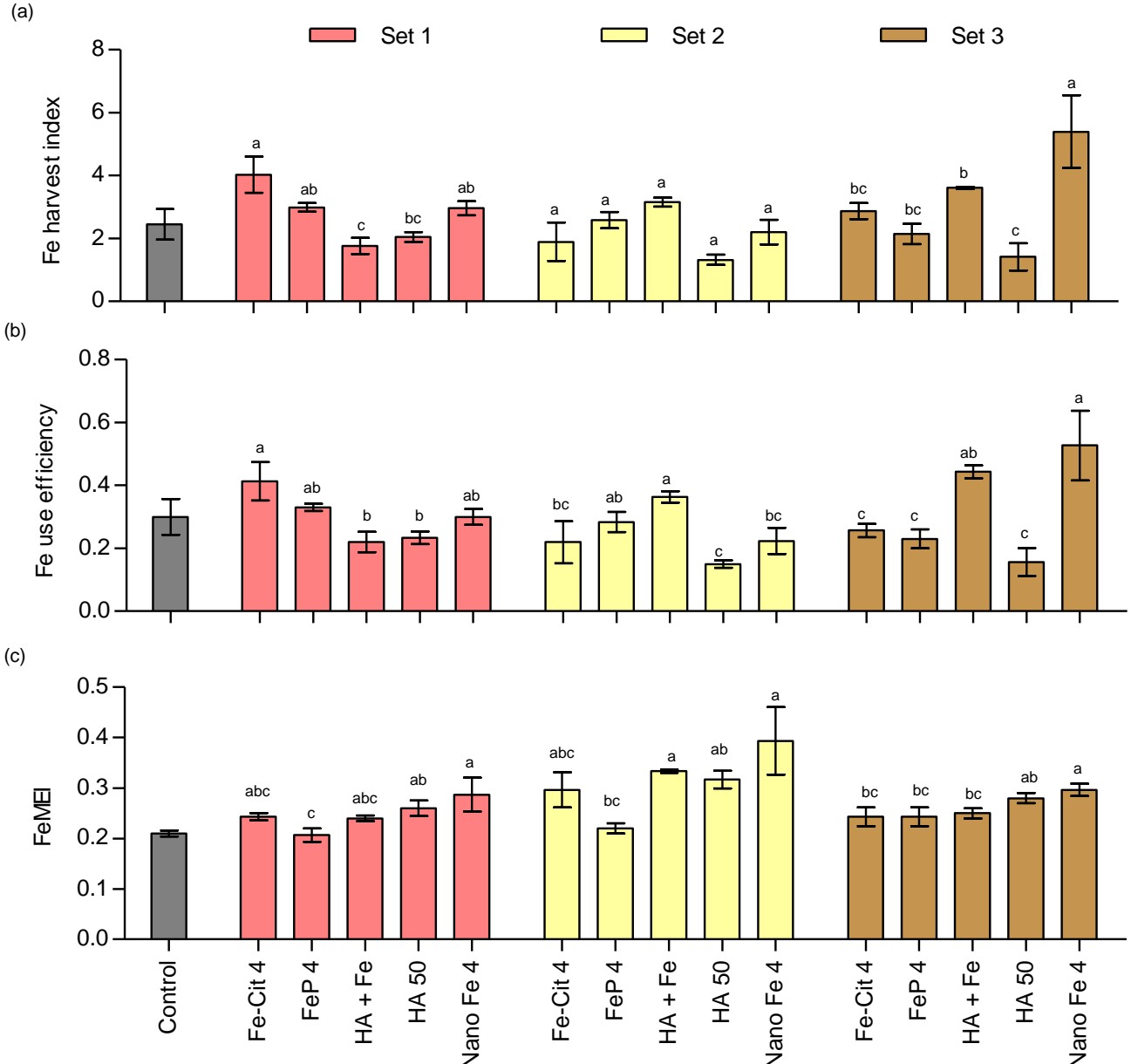

**Figure 9.** Influence of foliar application of Fe compounds on Fe indices in soybean: (**a**) Fe harvest index, (**b**) Fe use efficiency, and (**c**) Fe mobilization efficiency index (FeMEI). Data correspond to means $\pm$ SEm ($n = 5$). The plants of set 1 and set 2 received single sprays in the flowering and pod-filling stages, respectively, while set 3 plants were sprayed in both growth stages. Each set was analyzed separately using a one-way ANOVA, and the least significant difference was calculated. Means with same letter were not significantly different at $p < 0.05$. Details of the two-way ANOVA are presented in Table S2.

## 4. Discussion

Improving the Fe contents in food crops is an important challenge globally due to the high prevalence of Fe deficiency in humans. Our study demonstrates that the foliar application of Fe not only improved the Fe status of rice, soybean, and lettuce but also increased growth and yield. As suggested in earlier reports, we proved experimentally that the number and timing of foliar sprays influenced plant growth and yield in different crops. Our results showed that the maximum increases in the grain Fe concentrations and contents in both rice and soybean were achieved with three and two sprays, respectively. The growth stages of the plants at which the foliar application was performed greatly influenced the grain yield. In rice, single sprays during anthesis (set 2) or grain filling (set 3) exhibited the largest increases in grain yield, while in soybean spraying twice in the flowering and pod-filling stages (set 3) produced the maximum seed yield. Earlier reports showed that single sprays of different Fe formulations were sufficient for improving the growth and yield of rice [43], wheat, and soybean [23]. The flower yield and essential oil percentage in marigold (*Calendula officinalis*) achieved their maximum potential after a single spray of a 1000 ppm solution of nano-Fe during stem elongation compared to foliar application during flowering [44]. The foliar application of nano-Fe in the tillering and anthesis stages significantly improved growth and grain yield in wheat [26], while multiple sprays (>2 sprays during different developmental stages) were required to improve growth and yield in strawberry (*Frageria vesca*) [45] and maize [46].

Compared to other Fe compounds, the foliar application of humic acid (HA) with the addition of Fe (HA+Fe) considerably boosted the overall growth of all three crops, leading to enhanced biomass production in lettuce and significant improvements in the biomass and yield attributes in rice and soybean (Figures 2–4). As HA is an organic chelator, it forms complexes with nutrients (here, Fe) and increases their membrane permeability and solubility, resulting in increased nutrient uptake and metabolism [34,47]. Previous studies showed that the foliar application of HA has encouraging effects on plant growth and yield in several crops [6,34,48–50], but very few reported the combined effect of HA and Fe in crop plants. The application of HA significantly improved nitrate uptake by modifying root morphology and altering the expression of plasma membrane $H^+$-ATPase and nitrate transporters, resulting in greater root and shoot biomass in rice after different doses of HA applied in hydroponic solutions [51]. Further, Cimrin and Yilmaz [52] reported that soil-applied HA did not show any significant differences in biomass accumulation or phosphorus uptake in lettuce at lower doses (100 and 200 kg ha$^{-1}$), but a higher dose (300 kg ha$^{-1}$) increased the nitrogen content in shoots. This is in agreement with our results showing that a foliar application of HA+Fe significantly improved the dry matter percentages compared to the other treatments in lettuce plants, indicating that HA+Fe not only stimulates plant growth but also increases nutrient uptake (Figure 4b).

Fe foliar feeding increases the shoot Fe concentration, but its mobilization towards grains or seeds was greatly influenced by the type of Fe formulation, the number of sprays, and the plant growth stage. Our results indicated that if there was a low concentration of Fe in the source (the foliage of set 1), plants showed higher Fe mobilization efficiency, but the total amount of Fe mobilized toward grain was low. However, if the concentration of source Fe was increased by supplying exogenous Fe (foliage of set 5). Plant Fe mobilization efficiency decreased compared to set 1, but the total amount of Fe mobilized toward the grain was high. The results also revealed that the foliar application of nano-Fe-4 increased the Fe concentration in shoots/stovers and grain in both rice and soybean crops compared to other treatments (Table S4). Earlier studies also found that nano-Fe sprayed on crops such as cowpea (*Vigna unguiculata*), peanuts (*Arachis hypogea*), wheat, and soybean exhibited the highest grain Fe concentrations compared to other Fe fertilizers [6,53–55]. Compared to chelators and bulk Fe complexes, nano Fe showed better uptake due to its higher surface area to volume ratio [56,57]. Further, Bastani et al. [33] compared the absorption and mobilization of foliage-applied bulk Fe complexes and nano-Fe in tobacco plants and

concluded that nano-Fe exhibited greater mobility in the xylem and phloem, whereas the bulk Fe was largely retained in the foliage.

Our results showed a significant positive association between the Fe contents in grain and straw/stovers in both crops (Figures 5 and 6). We also observed that a high concentration of Fe in the foliage would not necessarily result in a higher grain Fe concentration because foliar-applied Fe was mostly retained by the foliage in the majority of Fe treatments. Previous studies supported the fact that, usually, most of the foliage-applied Fe chelates and bulk Fe complexes were retained in the foliage [33,55]. An exception to this was HA+Fe in both rice and soybean crops, as we found a significantly lower straw/stover Fe contents compared to other Fe treatments, thereby indicating higher mobilization of Fe from the foliage towards grains. Further, our results also proved that HA+Fe application significantly increased the Fe contents in lettuce leaves and the seeds/grains of soybean and rice compared to other treatments (Figures 5–7). Earlier studies have shown that the foliar application of either HA alone or in combination with Fe increased the uptake of nutrients in different crops [58–60]. Katkat et al. [34] reported a significant positive effect of the foliar application of HA on the uptake of both macronutrients (nitrogen, phosphorus, potassium, calcium, and magnesium) and micronutrients (zinc, Fe, copper, and manganese) in wheat plants grown in calcareous soil conditions. It was suggested that HA improves nutrient absorption from the foliage due to increased plasma membrane permeability, along with improved nutrient uptake through the roots due to improved root system development [50]. Very recently, Turan et al. [61] reported that nano-Fe and conventional Fe, along with HA, applied on spinach (*Spinacia oleracea*) improved growth and nutrient uptake significantly in comparison to control. Furthermore, they found no statistically significant difference between nano or conventional Fe as an HA supplement. The findings of this study strongly suggest that combining Fe with HA augment its effect on crop plants compared to foliar applications of HA or Fe formulations alone.

## 5. Conclusions

The foliar application of Fe fertilizers is an efficient and cost-effective way to increase the bioavailability of Fe in the edible parts of crops. Our results from three crops revealed that foliar Fe application not only improved the Fe status of crops but also enhanced their yields if the right Fe compounds were applied during the right growth stages. Further, HA+Fe produced the highest grain yields in both rice and soybean and the highest shoot biomass in lettuce. In both rice and soybean crops, seed Fe contents increased as the number of foliar sprays increased. However, the grain yield did not show any relationship with the number of sprays. This study strongly suggests that the foliar application of Fe along with HA enhances grain and biomass yields in rice and soybean as well as the dry matter content in lettuce, while nano Fe increases the Fe concentration in the seeds of rice and soybean crops.

**Supplementary Materials:** The following supporting information can be downloaded at https://www.mdpi.com/article/10.3390/agriculture13010132/s1, Table S1: Properties of soil used to grow rice, soybean, and lettuce crops in pots under natural environmental conditions. Soil sampling was performed before transplanting/sowing the crops. Table S2: Analysis of variance (ANOVA) for phenotypic traits of rice plants grown in pots under natural environmental conditions with foliar application of different Fe compounds. Table S3: Analysis of variance (ANOVA) for phenotypic traits of soybean plants grown in pots under natural environmental conditions with foliar application of different Fe compounds. Table S4: Effects of foliar treatments of various Fe compounds, applied in different growth stages, on Fe concentrations of rice and soybean plants grown in pots under natural environmental conditions.

**Author Contributions:** Conceptualization, R.P. and P.S.B.; methodology, S.S. and R.P.; formal analysis, S.S.; investigation, S.S.; resources, P.S.B.; writing—original draft preparation, S.S.; writing—review and editing, R.P., P.S.B. and N.A.; supervision, R.P.; project administration, R.P.; funding acquisition, R.P. All authors have read and agreed to the published version of the manuscript.

**Funding:** This research was funded by the Virtual Fertilizer Research Center, Washington, DC, USA, grant number 02838/14.

**Institutional Review Board Statement:** Not applicable.

**Data Availability Statement:** Not applicable.

**Conflicts of Interest:** The authors declare no conflict of interest.

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
