# Peer review of "Foliar Application of Humic Acid with Fe Supplement Improved Rice, Soybean, and Lettuce Iron Fortification"

_agriculture, doi:10.3390/agriculture13010132_

Round 1

Reviewer 1 Report

Dear Authors,

Thank you for the manuscript draft entitled "Foliar Application of Humic Acid with Fe Supplement Improved Rice, Soybean and Lettuce Iron-Fortification".

 My comments, suggestions, and views on this article are as follows:

 TITLE

Comment(s): The title is comprised of the study process in full, its' purpose, focus and area of interest.

Suggestion(s): -

 GRAPHICAL ABSTRACT (Optional)

Suggestion(s): the authors may consider submitting graphical abstract. Suggested contents for the graphical abstract: i) study overview/ Purpose of the study, and/or ii) Summary of Key findings for all objectives.

 ABSTRACT

Suggestion(s): Authors may wish to consider adding a brief study background / issues and/or problem statements - one / two sentence(s) and recommendations and/or further directions (further research / policy change and implementation / practise change) to the abstract. But note the instruction for authors.

 KEYWORDS

Comment(s): Good keywords in terms of number of words/ terms provided, brevity and do reflect the study.

Suggestion(s): - consider removing Latin names

 INTRODUCTION

Comment(s): This section has a good structure, cohesion and clarity. It includes the background of the study, issues, knowledge gap and practice deficiency, study goals, and research significance. This section also provides a relevant literature, an introduction to the methods used in this study.

Suggestion(s): -

 METHODS/ MATERIALS

Suggestion(s):

- Table S1: I suggest specifying the quality of the soil in accordance with the WRB (World Reference Base for Soil Resources).

- It was pot experiment I suggest give the contents of absorbable ingredients in mg∙kg-1 of soil.

- How optimal conditions for growth and development of soybeans were ensured (the length of the root system reaches 1.5 m and moisture conditions for rice).

 RESULTS, DISCUSSION

Suggestion(s): -

 CONCLUSION

Suggestion(s): -

 OVERALL: This is a good paper in terms of i) issues highlighted, ii) previous studies reviewed, iii) methods used, iv) findings and discussion, v) contributions to practice and body of knowledge, as well as.

Thanks  and best regards.

Author Response

Comment 1: The title is comprised of the study process in full, its' purpose, focus and area of interest.

Reply: Thank you for your inspiring words

Comment 2:  GRAPHICAL ABSTRACT (Optional)

Suggestion(s): the authors may consider submitting graphical abstract. Suggested contents for the graphical abstract: i) study overview/ Purpose of the study, and/or ii) Summary of Key findings for all objectives.

Reply:  Figure 1 can be used as graphical abstract.

Comment 3: ABSTRACT: Authors may wish to consider adding a brief study background / issues and/or problem statements - one / two sentence(s) and recommendations and/or further directions (further research / policy change and implementation / practise change) to the abstract. But note the instruction for authors.

Reply: Added

Comment 4:  KEYWORDS: Good keywords in terms of number of words/ terms provided, brevity and do reflect the study. Consider removing Latin names.

Reply: As reviewer suggestion, Latin name removed from the keywords.

 Comment 5:  INTRODUCTION: This section has a good structure, cohesion and clarity. It includes the background of the study, issues, knowledge gap and practice deficiency, study goals, and research significance. This section also provides a relevant literature, an introduction to the methods used in this study.

Reply: Thank you for your inspiring words.

 Comment 6:  Table S1: I suggest specifying the quality of the soil in accordance with the WRB (World Reference Base for Soil Resources).

Reply: We have mentioned the soil texture as sandy loam and other traits in the Table S1, however, we did not understand the ‘quality of soil’ as per WRBSR. Moreover, the experiment was conducted in pots and the objective was to identify the best foliar Fe compound and stage of crop growth, so we suppose that this information (specific name based on WRB) would matter much.

 Comment 7: It was pot experiment I suggest give the contents of absorbable ingredients in mg∙kg-1 of soil.

Reply: We have mentioned the recommended dose of NPK, however, we also calculated the amount of fertilizer as urea, single super phosphate, and muriate of potash that was applied (in g per pot) to each crop.

 Comment 8: How optimal conditions for growth and development of soybeans were ensured (the length of the root system reaches 1.5 m and moisture conditions for rice).

Reply: All three crops grow in India during their respective growing seasons to ensure optimal conditions for all three crops. Yes, it is a valid concern that the pot height is insufficient to accommodate the entire length of the rice and soybean roots. Nevertheless, pot experiments are conducted throughout the world in the same way by keeping a ‘control’ for each crop and comparing it with treated ones. Moreover, if there is a deficiency or nutrient (say P) or deficit water then the root growth is more but since our plants were supplied with sufficient water and nutrients, the root growth was enough to accommodate in the pots.

 Comment 9: OVERALL: This is a good paper in terms of i) issues highlighted, ii) previous studies reviewed, iii) methods used, iv) findings and discussion, v) contributions to practice and body of knowledge, as well as.

Reply: We thank you from the bottom of our hearts for your inspiring words for the manuscript, which will definitely boost our confidence.

Reviewer 2 Report

The study was a simple pot study on the foliar application of Fe on three crops.  The experimental design and statistics were well thought out.  However, there are missing elements from the introduction and discussion that setup the reader to better understand the importance of this research.  Relatively minor additions to these two sections and editing for grammar would improve this document.  Specific edits/recommendations are listed below:

Line 49- 51: Explain in more detail what strategy I and strategy II are and how they help the plant.

Line 54: delete “such as”

Line 58: colloquial language, use a different word for “hitch”

Lin 55-68: paragraph starts to lose focus. Reword for clarity.

Line 67: explain how increasing Fe is sustainable.  The following paragraph describes the process of folair application. 

Line 104-113: the gap of knowledge you are addressing is not clear in the last paragraph of the introduction.  Reword for clarity.

Line 117: delete “namely”

What was initial pH   the soil and how was it maintained if tap water was used for irrigation?

Was rice flooded?  If not flooded what was soil moisture of the plants during heading and grain fill?

Check grammar (line 140).

The text for the explanation of the sets in the foliar treatment is confusing.  Consider adding a table or figure to illustrate what you did.

Line 153: forenoon?  What time is that? If important explain why.

Consider renaming section 2.3 “Plant traits” or “phenotypes” observations implies only qualitative measurements.

Lines 163-167: reword for clarity.  The usage of “such as” suggests that you didn’t list all the traits measured.  In this section all traits measured need to be listed and as well as how they were measured. (i.e. test weight in what units?)

Line 168: what standard protocol?  Cite. Where were the chemicals sourced from?

Line 180: reword for clarity

Line 186: which protocol above.  Multiple are mentioned.

Figure 1. Is this for rice alone or all crops together?  Remove copyright symbol in description and replace with (c). 

A mixed model analysis of Variance means table would be easier to read than figure 1.  It’s hard to tell what is significant compared to controls.  

Table S4: change units so data isn’t presented the hundreds place. Too many zeros make it hard to read.

Figure 3.  Why is graph in grey scale where 1 & 2 are in color?   If only one set, then use one color for uniformity.

Line 444-446: reword for clarity.

Lines: 469 – 476 are discussing a similar topic to lines 427 – 443.  Combine topics/ paragraphs if possible. 

Mobility of Fe is discussed, but no introduction to the topic is made.  Is it mobile without HA, what percent?

Does the foliar application of Fe cause these three plants to reach the human requirement of Fe?  What percent? Is there any economical implications with adding 1 vs 3 foliar applications of Fe? Does it differ for each species?   What is impact of your research to producers around the world?

Author Response

Reviewer: 2

Major comment: The study was a simple pot study on the foliar application of Fe on three crops.  The experimental design and statistics were well thought out.  However, there are missing elements from the introduction and discussion that setup the reader to better understand the importance of this research.  Relatively minor additions to these two sections and editing for grammar would improve this document.  Specific edits/recommendations are listed below:

Reply: We are very thankful to the reviewer who provided their valuable time to improve our manuscript. All the suggestions have been incorporated into the manuscript.  

Comment 1: Line 49- 51: Explain in more detail what strategy I and strategy II are and how they help the plant.

Reply: The details of Strategies I and II have been added to the Introduction, as suggested by the reviewers (Line 53-59).

Comment 2: Line 54: delete “such as”

Reply: Deleted as reviewer suggested

Comment 3: Line 58: colloquial language, use a different word for “hitch”

Reply: As per reviewer suggestions “hitch” word replace by “challenges”

Comment 4: Lin 55-68: paragraph starts to lose focus. Reword for clarity.

Reply: This paragraph is reworded to bring out the advantages of agronomic approach over the other two, conventional breeding and transgenic approaches.

Comment 5: Line 67: explain how increasing Fe is sustainable.  The following paragraph describes the process of foliar application.

Reply: Yes, it was a wrong statement, foliar application does not come under sustainable agronomic practices. The word "sustainable" is removed from the sentence.  

Comment 6: Line 104-113: the gap of knowledge you are addressing is not clear in the last paragraph of the introduction.  Reword for clarity.

Reply: The paragraph is reworded to bring out the clarity in the knowledge gap that has been addressed in our study.

Comment 7: Line 117: delete “namely”

Reply: Deleted as reviewer suggested

Comment 8: What was initial pH of the soil and how was it maintained if tap water was used for irrigation?

Reply: The initial pH of the soil was measured as 8.4 (Table S1). The pH of the soil was not maintained; however, soil from the same pool was used for growing control plants as well.

Comment 9: Was rice flooded?  If not flooded what was soil moisture of the plants during heading and grain fill?

Reply: Yes, soil was flooded till anthesis stage, the water level being 4-5 cm above the soil surface. However, during grain filling, the soil was kept sufficiently moist, not flooded. We did not measure soil moisture during any growth because our experiment does not relate to water stress.

Comment 10: Check grammar (line 140).

Reply: Corrected

Comment 11: The text for the explanation of the sets in the foliar treatment is confusing.  Consider adding a table or figure to illustrate what you did.

Reply: Thank you, it is a good suggestion, so we have provided a figure which describes the layout of foliar applications.

Comment 12: Line 153: forenoon?  What time is that? If important explain why.

Reply: The plants were sprayed with Fe solution in the forenoon (before 10 am), because during this time, the humidity is higher and leaves are in a state of full turgor, which leads to maximum absorption of nutrients from foliage. This is added in the text.

Comment 13: Consider renaming section 2.3 “Plant traits” or “phenotypes” observations implies only qualitative measurements.

Reply: The name of section 2.3 changed to “physiological traits and tissue nutrients analysis”

Comment 14: Lines 163-167: reword for clarity.  The usage of “such as” suggests that you didn’t list all the traits measured.  In this section all traits measured need to be listed and as well as how they were measured. (i.e., test weight in what units?)

Reply: Thanks for the important comment will improve the material methods section. We have replaced "such as" by "including" (line 184). The unit of test weight is added.

Comment 15: Line 168: what standard protocol?  Cite. Where were the chemicals sourced from?

Reply: The chemical source and reference are added.

Comment 16: Line 180: reword for clarity

Reply: Reworded.

Comment 17: Line 186: which protocol above.  Multiple are mentioned.

Reply: Corrected. Same protocol for Fe estimation was used for lettuce as mentioned for rice and soybean.

Comment 18: Figure 1. Is this for rice alone or all crops together?  Remove copyright symbol in description and replace with (c). 

Reply: No, Figure 2 (earlier Figure 1) depicted the biomass and yield attributes of rice crop. The copyright symbol was removed and replaced with correct serial number of Figure 2.

Comment 19: A mixed model analysis of Variance means table would be easier to read than figure 1.  It’s hard to tell what is significant compared to controls.  

Reply: Each set of data was analysed separately by one-way ANOVA, and the means with the same letter showed no significant difference at P<0.05. While for detecting the interaction between different sets and treatments, the details of a two-way ANOVA were mentioned in Supplementary Table 2.

Comment 20: Table S4: change units so data isn’t presented the hundreds place. Too many zeros make it hard to read.

Reply: It is a valuable comment. The unit of data presented in Table S4 has been changed from mg g-1 to µg g-1

Comment 21: Figure 3.  Why is graph in grey scale whereas 1 & 2 are in colour?   If only one set, then use one color for uniformity.

Reply: Nice suggestion indeed, all grayscale graphs have been replaced by colour graphs.   

Comment 22: Line 444-446: reword for clarity.

Reply: Corrected.

Comment 23: Lines: 469 – 476 are discussing a similar topic to lines 427 – 443.  Combine topics/ paragraphs if possible. 

Reply: We respect the reviewer's opinion, but we feel that combining these paragraphs will disturb the reader's flow. We discussed the results here trait-wise. The first paragraph (lines 427-443) discussed the effect of Fe treatment on biomass and yield, and concluded that HA+Fe showed the best results among treatments in all three crops. While the second paragraph (Lines 469-476) discussed the correlation between Fe concentration in grain and straw/stover in these crops which is explained on the basis of Fe mobilization from straw/stover towards grain. 

Comment 24: Mobility of Fe is discussed, but no introduction to the topic is made.  Is it mobile without HA, what percent?

Reply: We have introduced HA in the Introduction.

Comment 25: Does the foliar application of Fe cause these three plants to reach the human requirement of Fe?  What percent? Is there any economical implications with adding 1 vs 3 foliar applications of Fe? Does it differ for each species?   What is impact of your research to producers around the world?

Reply: Yes, foliar application of Fe fertilisers significantly (8-66%) increased the Fe concentration in the edible part of all three crops, however, the percent increase in Fe concentration depends on the number of sprays, plant growth stages, and type of Fe compound used. We found that the number of sprays is directly proportional to the increase in Fe concentration in the grain/seed of rice and soybean; however, a higher number of sprays decreased plant growth and yield, and excessive foliar application may cause Fe toxicity in the plants. Our results showed that humic acid with Fe supplement improved overall plant growth, increased yield, and boosted Fe concentration in the edible parts of all three crops. Yes, the number of foliar sprays does have economic implications, and three foliar sprays would cost more although we did not calculate the cost-benefit. The findings of this study will have a positive impact as it provides enough evidence of Fe-biofortification through the foliar application, which is a simple method and can be adopted in countries with less available means to address the Fe deficiency in humans worldwide.